Evaluation of traditional and bootstrapped methods for assessing data-poor fisheries: a case study on tropical seabob shrimp (Xiphopenaeus kroyeri) with an improved length-based mortality estimation method

de Barros Matheus mbarros@disl.org 1
Oliveira-Filho Ronaldo 2 3
Aschenbrenner Alexandre 2 3
Hostim-Silva Mauricio 2 3
Chiquieri Julien 3
Schwamborn Ralf 4
1 School of Aquatic and Fishery Sciences, University of Washington , Seattle , WA , United States of America
2 Departamento de Ciências Biológicas, Universidade Federal do Espírito Santo , Vitória , Espírito Santo , Brazil
3 Departamento de Ciências Agrárias e Biológicas, Universidade Federal do Espírito Santo , São Mateus , Espírito Santo , Brazil
4 Departamento de Oceanografia, Universidade Federal de Pernambuco , Recife , Pernambuco , Brazil
Norris Darren
Electronic publication date: 2024 Nov 14
Publication date: 2024
Volume: 12
Electronic Location ID: e18397
Received 2023 Jan 18; Accepted 2024 Oct 4
Copyright: ©2024 de Barros et al.
Copyright year: 2024
Copyright holder: de Barros et al.
License: This is an open access article distributed under the terms of the Creative Commons Attribution License, which permits unrestricted use, distribution, reproduction and adaptation in any medium and for any purpose provided that it is properly attributed. For attribution, the original author(s), title, publication source (PeerJ) and either DOI or URL of the article must be cited.
License URL: https://creativecommons.org/licenses/by/4.0/

Keywords: Penaeidae, Crustacean, Stock evaluation, Powell-Wetheral, ELEFAN, Length-frequency analysis, Catch curve, Mortality, Growth, Von Bertalanffy

Funding: Renova Foundation 001/2018 This work was supported by the Renova Foundation (001/2018). The funders had no role in study design, data collection and analysis, decision to publish, or preparation of the manuscript.

==============================
Background

Unrealistic model assumptions or improper quantitative methods reduce the reliability of data-limited fisheries assessments. Here, we evaluate how traditional length-based methods perform in estimating growth and mortality parameters in comparison with unconstrained bootstrapped methods, based on a virtual population and a case study of seabob shrimp (Xiphopenaeus kroyeri, Heller, 1862).

Methods

Size data were obtained for 5,725 seabob shrimp caught in four distinct fishing grounds in the Southwestern Atlantic. Also, a synthetic population with known parameter values was simulated. These datasets were analyzed using different length-based methods: the traditional Powell-Wetheral plot method and novel bootstrapped methods.

Results

Analysis with bootstrapped ELEFAN (fishboot package) resulted in considerably lower estimates for asymptotic size (L∞), instantaneous growth rate (K), total mortalities (Z) and Z/K values compared to traditional methods. These parameters were highly influenced by L∞ estimates, which exhibited median values far below maximum lengths for all samples. Contrastingly, traditional methods (PW method and Lmax approach) resulted in much larger L∞ estimates, with average bias >70%. This caused multiplicative errors when estimating both Z and Z/K, with an astonishing average bias of roughly 200%, with deleterious consequences for stock assessment and management. We also present an improved version of the length-converted catch-curve method (the iLCCC) that allows for populations with L∞ > Lmax and propagates the uncertainty in growth parameters into mortality estimates. Our results highlight the importance of unbiased growth estimates to robustly evaluate mortality rates, with significant implications for length-based assessments of data-poor stocks. Thus, we underscore the call for standardized, unconstrained use of fishboot routines.

Introduction

Globally, fisheries provide employment to hundreds of millions of people, generate hundreds of billions in trade revenue, and are the economic, social and cultural backbone of numerous societies, especially in developing countries (FAO, 2020). Yet, most fisheries nowadays face an inevitable crisis due to chronic stock deterioration, which is largely attributed to the lack of proper assessment and management. Roughly half of all ecologically and economically important fish stocks around the world are harvested by small-scale, artisanal, or semi-artisanal fisheries in developing countries. These fisheries are known to employ about 90% of the world’s fishers (Jacquet & Pauly, 2008). Despite their importance, these fisheries often do not benefit from formal, data-rich coordinated assessments due to insufficient funding or government policies (Jacquet & Pauly, 2008; Wakamatsu & Wakamatsu, 2017). Consequently, they frequently receive inadequate assessments, or none at all. Recognizing their significance, FAO has recently classified data-limited, small-scale fisheries as priorities for future assessment and management programs (FAO, 2022).

Small-scale fisheries often lack important data such as catch-at-age, abundance time series, and precise life-history information, hindering the implementation of more complex stock assessment models (Carruthers et al., 2014; Costello et al., 2012). Therefore, alternative modeling tools have been developed to assess data-limited stocks (Carruthers et al., 2014). Length-frequency analysis (LFA, Petersen, 1891) is one of the most popular indirect methods for fitting a von Bertalanffy growth function (VBGF, Von Bertalanffy, 1938) to length data and obtain basic demographic parameters, particularly for data-poor tropical fisheries (Sparre & Venema, 1992). LFA can estimate growth, mortality, recruitment patterns, and provide the basis to derive biological reference points for management (Sparre & Venema, 1992). For crustacean stocks (e.g., shrimps, crabs and lobsters), LFA is the only cost-effective method available, as these organisms do not have hard structures for aging (e.g., scales, otoliths, vertebrae, etc). While tagging (mark-recapture) is a possible direct method for estimating body growth in crustaceans, it is often complex, laborious and cost-intensive (e.g., Schwamborn & Schwamborn, 2021), and thus rarely used. Conversely, LFA requires length data only and remains the most popular method for the quantitative evaluation of data-poor fisheries.

Among the techniques used to conduct LFA with search algorithms, the ELEFAN method (Electronic Length Frequency Analysis, Pauly & David, 1981) is well-established and widely used to estimate body growth from length-frequency distributions (LFDs). ELEFAN algorithms perform modal progression analysis to find the VBGF curve that best fits the monthly LFDs by maximizing the number of peaks the curve intersects. Recently, new versions of ELEFAN (“ELEFAN in R”, “TropFishR” and “fishboot”) have been developed within the R environment (Pauly, 2013; Mildenberger, Taylor & Wolff, 2017; Schwamborn, Mildenberger & Taylor, 2019). The “TropFishR” package (Mildenberger, Taylor & Wolff, 2017) enhances the original ELEFAN search algorithm with more efficient computational optimization techniques to improve parameter estimation, such as genetic and simulated annealing, building on the original “Compleat ELEFAN” (Gayanilo Jr & Pauly, 1987) and “FISAT” II (Gayanilo Jr, Sparre & Pauly, 1996; Gayanilo & Sparre, 2005) software packages. The recent “fishboot” package (Schwamborn, Mildenberger & Taylor, 2019) extends “TropFishR” by introducing a new, bootstrapped framework of algorithms that incorporates multidimensional uncertainty and, most importantly, enabling a shift from a biased single-fit approach to more robust and reliable evaluations of the multidimensional search space of growth parameters.

Despite recent critiques by Schwamborn (2018), traditional methods such as the Powell-Wetheral (PW) plot (Wetherall, Polovina & Ralston, 1987) remain widely used in recent fisheries assessments. This method typically fixes one single value for the asymptotic length L∞ prior to running ELEFAN to estimate the instantaneous growth rate (K) separately (Ayo-Olalusi & Abeke Ayoade, 2018; De Los Ríos et al., 2019; Livore, Arrighetti & Penchaszadeh, 2021; Lucena-Frédou et al., 2017). According to Schwamborn (2018), this traditional approach is unreliable in estimating L∞. These issues are particularly true if there is intra-cohort variability in life-history traits and changing selectivity dynamics, which are common in most fisheries. The most serious flaw in the PW method arises from a spurious autocorrelation between the two axes in the regression analysis, leading to a consistently significant trend and overconfidence in the L∞ estimate regardless of data structure (Schwamborn, 2018).

Schwamborn (2018) recommended the simultaneous estimation of L∞ and K parameters rather than fixing of L∞ a priori. The R package “fishboot” (Schwamborn, Mildenberger & Taylor, 2019) provides a statistical optimization framework that fits growth curves to length frequency data using bootstrapping to account for observation/process uncertainty, thereby providing estimates of uncertainty for growth parameters. In this study, we aimed to (1) test the performance of a new method to estimate total mortality rates based on length composition, named the “improved length-converted catch curve”, (2) compare the parameter estimation performance of the fishboot package with the traditional PW method in simulated populations with known life-history traits; and (3) evaluate the relative performance of the two methods using a real-world scenario: seabob shrimp Xiphopenaeus kroyeri in the southwestern Atlantic.

Materials & Methods

The improved length-converted catch curve (iLCCC)

The standard method to estimate total mortality (Z) from size-frequency data is the length-converted catch curve (LCCC, Baranov, 1918; Pauly, 1983; Pauly, 1984a; Pauly, 1984b). This method transforms length classes into relative ages according to input growth parameters. A considerable limitation of the LCCC is its inapplicability when the asymptotic length (L∞) is lower than the maximum size in the sample. This limitation essentially excludes populations with lower mortality/growth (Z/K) ratios, as these typically exhibit higher frequencies of large-sized individuals due to lower relative harvest pressure (Schwamborn, 2018). To address this issue, we developed a new algorithm, the iLCCC (“improved length-converted catch-curve”, de Barros, 2022), which automatically excludes size classes that are larger than the estimated L∞ and performs bootstrapping to propagate uncertainty from growth parameters into the resulting mortality estimate. The iLCCC is based on the following framework:

First, we assume numbers-at-age (Nt) decrease over time according to an instantaneous rate of total mortality (Z): (1) Nt=Nt−1e−Zt

(2) lnNt=lnNt−1−Zt

We then transform the length composition to relative ages (aL) using growth parameter estimates from the von Bertalanffy growth function (Essington, Kitchell & Walters , 2001; Von Bertalanffy, 1957) as follows: (3) aL=t0−1Kln1−LL∞

where aL is the relative age (the estimated age for the size class L), t 0 (year−1) is the theoretical age when length equals zero, which determines the horizontal position of the growth curve (Kirkwood, 1983; Von Bertalanffy, 1957), K (year−1) is the instantaneous growth rate, and L∞(mm) is the theoretical asymptotic length. The instantaneous rate of mortality (Z, year−1) is then estimated as the slope of a simple linear regression fitted to the decline in numbers-at-age through time, using relative age (aL) as a proxy (Eqs. (1) and (2)).

The major advantage of the iLCCC lies in the use of bootstrapping to resample relative age estimates based on the uncertainty in growth parameters. Specifically, we employ a “two-step” bootstrap (Schwamborn & Schwamborn, 2021), where Z is calculated multiple times by iterating through randomly generated samples of K, L∞, and t0, as well as by resampling the input regression data. Despite its relative simplicity and similarity to the traditional length-converted catch-curve, we argue this method represents a substantial improvement because it essentially propagates the uncertainty in life-history traits to the resulting mortality estimate. This is often not incorporated in length-based assessments, possibly resulting in underestimates in the underlying uncertainty in mortality rates.

Simulation study

We first conducted a simulation study to (1) determine the reliability of the iLCCC method in estimating total mortality rates from length composition data, and (2) compare growth and mortality estimates between the fishboot package and the traditional PW approach, using known “true” parameter values as baseline. The synthetic population was simulated using the “fishdynr” package (Taylor & Mildenberger, 2017) with pre-determined growth and mortality rates (Table 1). A year of length frequency data was extracted from the model. This data was then used to estimate growth and mortality rates using bootstrapped ELEFAN and the PW method, as detailed in the following sections.

Table 1 Parameters used to simulate the synthetic population.

Parameter	Value	CV	
K (year −1)	0.5	0.1	
L ∞ (mm)	200	0.1	
t_anchor	0	NA	
Z (year −1)	0.75	0.1	
Notes.

Means and coefficients of variation (CV) for the parameters used to simulate the virtual population. K stands for the instantaneous growth rate, L∞ is the theoretical asymptotic length, t_anchor is used to anchor the start of the growth curve throughout the year, and Z is the instantaneous total mortality rate.

Real world scenario—study area and sampling

The seabob shrimp Xiphopenaeus kroyeri is a widely distributed penaeid inhabiting shallow coastal waters in the tropical and subtropical Western Atlantic. It is a highly popular target species for artisanal small-scale fisheries (Musiello-Fernandes, Zappes & Hostim-Silva, 2018; de Barros et al., 2022). Shrimp were sampled monthly from October 2018 to September 2019 at coastal sampling stations of four important shrimp fishing grounds in the Eastern Brazilian Marine Ecoregion (sensu Spalding et al., 2007) (Fig. 1). These areas included: Caravelas (CV), São Mateus (SM), Ipiranga (IP), and Rio Doce (RD). The Caravelas estuary encompasses the delta of the Peruípe River, which is 58 km long with a watershed of 4.780 km2, and includes the Cassurubá Extractive Reserve, a Marine Protected Area of extractive use. The São Mateus River, approximately 76 km long, has a basin that drains about 8.000 km2, formed by the confluence of the Cotaxé and Cricaré rivers (Condini et al., 2022). The Ipiranga River extends about 138 km long, as the southern extension of the Barra Seca River (Silva et al., 2018). The Rio Doce River runs for 850 km from the Mantiqueira and Espinhaço mountains to the Atlantic Ocean, with a watershed basin covering about 86.715 km2 (Schettini & Hatje, 2020).

Figure 1 Map of sampling locations.

Map of sampling regions created using QGIS.

Artisanal fisheries are an important source of income for local communities in the study areas. The coastal zones near the mouths of the Doce and Caravelas rivers are major fishing grounds, where seabob shrimp are exploited (Oliveira et al., 2020). Seabob shrimp samples were collected using trawl nets (13 mm mesh size in the net body and 15 mm in the cod end) for five minutes. Three replicate tows were conducted at four stations in each area at the beginning of each month, resulting in 48 (3 × 4 × 4) samples per campaign (Condini et al., 2022; Oliveira-Filho et al., 2023). After 12 monthly campaigns, this resulted in a total of 576 samples. Seabob shrimp were immediately euthanized in ice slurry and kept frozen laboratory processing, where a subsample of 30 individuals was randomly selected from each haul for measuring total length (TL), carapace length (CL) and wet weight (WW) (Fig. 2).

Figure 2 Size distributions for seabob shrimp collected at the four fishing grounds.

Size distributions for seabob shrimp collected at the four fishing grounds and summary statistics. The blue and red lines represent medians and its respective confidence intervals.

Assessing growth and mortality with traditional length-based methods

For both synthetic (simulated) and real-world scenarios, we initially estimated L∞ and Z/K for each sample using a “traditional” length-based approach, specifically the modified version of the PW-plot method (Pauly, 1986). This method relies on the partition of the size frequency distribution into size classes of equal intervals using a series of arbitrary cutoff lengths (Lc). The mean lengths of all fish within each i-th size class (Lmean[i]) are then calculated and used to fit a simple linear regression of the difference between the midpoints and cutoff lengths (Lmean - Lc) against Lc. The intercept (α) and slope (β) estimates from this regression are then used to calculate L∞ as L∞ = α/- β and Z/K as Z/K = β/(1- β).

In this framework, data points were automatically selected using standardized “gamma” selection (initial point: “mode + 10%”; last point: 80% of the maximum length, R code available at http://rpubs.com/rschwamborn/267465). Although this selection was arbitrary, we previously verified that other values for the last point yield similar results. The estimated L∞ values from the PW method were then used as fixed inputs for subsequent estimation of the instantaneous growth rate K through the “ELEFAN_GA_boot” function (Schwamborn, Mildenberger & Taylor, 2019) Then, total instantaneous mortality (Z) was estimated by the iLCCC method using these K and L∞ values as inputs.

Length-frequency analysis with fishboot

VBGF parameters (L∞, K) were estimated simultaneously using modern bootstrapped algorithms within the fishboot toolbox (Schwamborn, Mildenberger & Taylor, 2019). First, LFDs for each region were organized by sampling months with a bin size of four mm following the rule of thumb by Wang et al. (2020) where optimal bin size = 0.23 × Lmax0.6. These LFDs were then used as input data for the bootstrapped ELEFAN with a genetic search algorithm (ELEFAN_GA_boot) implemented in the “fishboot” R package (Schwamborn, Mildenberger & Taylor, 2018; Schwamborn, Mildenberger & Taylor, 2019), which builds on functions from the “TropFishR” R package (Mildenberger, Taylor & Wolff, 2017). This algorithm employs a complex genetic framework based on natural selection (Xiang et al., 2013) to improve the goodness-of-fit in the ELEFAN curve fitting process, initially applied to LFA within the “TropFishR” package (Mildenberger, Taylor & Wolff, 2017). It searches for optimal von Bertalanffy growth equation parameters by testing several combinations through multiple iterations until finding a local maximum (i.e., the curve that best performs with the given LFDs). For each study area, LFD data was resampled 500 times in the bootstrap routine using the ELEFAN_GA_boot algorithm in fishboot (Schwamborn, Mildenberger & Taylor, 2018). A very wide and unconstrained search space was used (K: 0.05 to 3 yr−1, L∞: 50 to 200 mm, t_anchor: 0–1), following standard procedures for unbiased parameter search (Schwamborn, Mildenberger & Taylor, 2019). This approach was considered more accurate as it is intrinsically unrestricted, and therefore assumed to represent the most likely population parameter distribution without potential artifacts from a priori defined narrow search space limits (Zhou et al., 2022). The moving average (MA) was set to a value of 11 since higher values improve modal detection when using intermediate bin sizes (Taylor & Mildenberger, 2017).

Natural mortality

Natural mortality rates (M) for seabob shrimp in the real-world scenario were estimated by the growth-based empirical method of Then et al. (2015) as follows: M=4.118K0.73L∞−0.33

where K is the instantaneous growth rate (yr−1), and L∞ is the theoretical asymptotic length in cm.

Comparing total mortality (Z) estimates fromtraditional and bootstrapped methods

Relative bias of the PW method (RB, in %) was calculated relative to the fishboot posterior distributions as following: RB%=Observation−Estimate/Observation×100.

For the simulation study, relative bias was calculated based on the true input parameters from the synthetic population (Table 1). For the seabob shrimp real-world scenario, fishboot outputs were considered to be “observations” and outputs from the PW model were considered as “estimates”. Subsequent mentions of “bias” in the results refer to relative bias outlined in this section.

For each parameter (L∞, K, Z, etc.), 95% confidence intervals (CI) were calculated from bootstrapped posterior distributions as the 0.25 and 97.5% quantiles.

Results

Simulation study

After 500 bootstrap iterations, the genetic optimization algorithm consistently converged to the “true” parameter values (K, L∞) used to simulate the population, with less than 5% of bias in the mean posterior distributions relative to the known parameters (Table 1), and relatively narrow uncertainties. Subsequent mortality estimates from the iLCCC using the estimated growth parameters to calculate relative age were also accurate, with less than 5% of bias (Fig. 3). Contrastingly, growth parameter and subsequent mortality estimates from the PW method showed significant bias, exceeding 10% for asympptotic length L∞, and almost 30% and 50% of bias for the instantaneous growth rate K and total mortality Z, respectively (Fig. 3).

Figure 3 Distributions of relative bias in parameter estimation for the synthetic populaiton between the two methods.

Means, quantiles, and standard deviations of relative bias on growth parameters and total mortality from the traditional PW method and bootstrapped ELEFAN for the synthetic population.

Real-world scenario

The median best-fit parameter estimates obtained from the fishboot analysis of seabob shrimp length compositions resulted in L∞ values considerably below the overall Lmax for all regions, with reasonably small confidence intervals (CI = 8%). Growth coefficients (K) showed a rather distinct behavior between samples, with high intra-cohort variability and wide distributions (ranges between 0.44 and 1.85, CI = 25%) for all regions. Most importantly, the best-fit L∞ was much smaller than Lmax for all samples, with only values larger than the third quantile of the posterior distributions of L∞ being larger than Lmax. This discrepancy precluded the use of the traditional LCCC method and led to the development of the iLCCC method (see methods section). Similarly to the growth parameter estimates, Z estimates displayed relatively large values, with wide 95% confidence intervals (Table 2).

Table 2 Growth and mortality parameter values.

Medians and 95% confidence intervals for growth and mortality parameters at each region from the fishboot model.

	SM	RD	IP	CV	
Parameter	Median	95% C.I.	Median	95% C.I.	Median	95% C.I.	Median	95% C.I.	
Linf (mm)	99.3	(87.6–104.9)	96.9	(94.4–102.1)	94.8	(89.3–112.9)	94.7	(83.8–102.2)	
K (year−1)	1.12	(0.37–1.27)	0.93	(0.38–0.98)	0.72	(0.4–1.06)	0.7	(0.47–1.38)	
Z (year−1)	1.25	(0.35–1.087)	0.83	(0.3–0.85)	0.65	(0.41–1.18)	0.67	(0.48–1.41)	
M (year−1)	1.11	(0.38–1.24)	0.93	(0.37–1.16)	0.74	(0.48–1.12)	0.71	(0.53–0.88)	
Z/K	1.1	(0.68–0.99)	0.89	(0.74–0.86)	0.9	(0.9–1.11)	0.95	(0.65–1.51)	

Growth/mortality (Z/K) ratios obtained using fishboot were very close to 1 and showed persistent multimodal distributions (Fig. 4 and Table 2). Further comparisons between the outputs of both methods suggested that the PW method is inappropriate to estimate L∞ and subsequent analyses, at least in populations with a low Z/K ratio. Mortality estimates were remarkably different (by more than four-fold) between the fishboot (ELEFAN_GA_boot and subsequent iLCCC) and traditional (PW-plot, ELEFAN_GA_boot and subsequent iLCCC) methods, with considerable between-region variability (Fig. 5). This resulted in an astonishing amount of bias in the mortality estimates from the traditional PW approach, likely due to a multiplicative effect from the bias in L∞. In fact, a bias of only 20% in L∞, which might be considered as “moderate”, can lead to more than 200% of bias on total mortality Z. The PW method, with such a huge amount of bias, may be considered an extremely biased or “erroneous” approach. (Fig. 6). Additional analysis revealed that the total mortality estimates resulting from the PW method are highly correlated with the bias in L∞. Consequently, bias in both total mortality and Z/K values appear to be highly dependent on the bias in L∞ (Fig. 7). Interestingly, natural mortality (M) for the four different areas were remarkably similar to or even greater than total mortality values (Table 2). While this would imply minimal fishing at the four fishing grounds, it is more likely that such estimates are invalid for the studied populations.

Figure 4 Parameter distributions for bootstrapped ELEFAN.

Density distributions for growth parameter estimates obtained from bootstrapped ELEFAN (asymptotic length Linf, growth rate K), mortality growth ratios (Z/K), and total mortality (Z) for seabob shrimp at each fishing ground.

Figure 5 Parameter distributions between the two methods.

Medians, quantitles, and standard deviations of total mortality (Z), mortality/growth ratios (Z/K), and asymptotic length (Linf) for the traditional PW method and bootstrapped ELEFAN at each fishing ground where seabob shrimp samples were obtained.

Figure 6 Bias in parameter estimation.

Medians, quantiles and standard deviations for biases in asymptotic length (Linf) and Z/K ratios for seabob shrimp at different regions. Green regions stands for “acceptable” bias, yellow stands for “moderate” bias, and red stands for “extreme” bias.

Figure 7 Relationships between errors on the estimation of different parameters.

Bias in the mortality/growth relationship increases exponentially with increasing bias in the asymptotic length.

Discussion

Our study reinforces that widely used traditional length-based methods are inappropriate as stock assessment tools, as natural populations are unlikely to follow its assumptions. In both simulated and real-world scenarios, the PW method highly overestimated the asymptotic length L∞ compared to the fishboot method, one of the most important growth parameters that directly influences subsequent growth and mortality estimates from catch curve analyses. Our results demonstrated that total mortality values derived from L∞ estimates using the PW method were hugely biased, with a deviation of more than 500% relative to the bootstrapped ELEFAN method (fishboot) with unconstrained search space. We also present an improved method for estimating mortality (iLCCC) that allows L∞ to be <Lmax during LCCC. This apparently minor improvement has important implications for stock assessment, as it allows the assessment of total mortality using LCCC in situations where Lmax is above L∞ (e.g., Type “A” populations sensu Schwamborn (2018), with Z/K far below 2 and high variability in L∞). Previously, the imperative for L∞ estimates to be above or equal to Lmax for mortality estimation with LCCC needed a priori constrained L∞ search limits during ELEFAN routines, potentially leading to severe bias in growth and mortality estimates. Our simulation study also indicates substantial accuracy of the iLCCC method when growth parameters are unbiased (Fig. 3).

Pitfalls in parameter estimation with traditional methods

According to the simulations conducted by Schwamborn (2018), the PW-plot routine considerably overestimates L∞ for type “A” populations (low mortality/growth relationships and maximum length larger than L∞). This was also observed in our study, where the simulated population showed very low Z/K values using fishboot, which appear to be strongly influenced by L∞ (Fig. 3 and Table 1). For our seabob shrimp data, the PW method consistently produced L∞ values that were much larger than the maximum length of the sample. Contrastingly, the best-fit L∞ values obtained using the fishboot method were much smaller than the maximum length for all samples (Table 1). This culminated in deviations larger than 50% between the results of both methods in most runs (see Fig. 5). Ideally, mathematical and statistical models attempting to describe the dynamics of natural processes should strive to predict system behaviors with a minimal number of parameters and reduced complexity. However, unrealistic and oversimplifying assumptions can lead to misleading overconfidence on the model’s ability to describe a process or estimate a crucial parameter (Collie et al., 2016; Haddon, 2011). This is the case with the PW routine: as stated by Schwamborn (2018), its assumptions—such as steady-state equilibrium conditions, no intra-cohort/seasonal variability in growth parameters, and no gear avoidance (perfectly-sampled populations)—are major sources of errors in estimating asymptotic size. Another oversimplifying assumption, present in widely referenced fisheries textbooks (e.g., Sparre & Venema, 1992), is that L∞ should always be above the maximum observed size. Traditional methods to estimate growth parameters through LFA such as the von Bertalanffy plot do not accept lengths that are larger than L∞ (e.g., “The argument of the logarithm…must be positive as the logarithm would otherwise not be defined. Thus, the von Bertalanffy plot cannot accept a length greater than L∞”, Sparre & Venema, 1992). This assumption (Lt >L∞) holds true only when there is little and homogenous variability in growth (not realistic), or when there is a high mortality/growth ratio (as in “dwarfed” populations, see Schwamborn & Moraes-Costa, 2019).

Overestimation of L∞ from the PW method can lead to considerable errors in estimating total mortality with the widely used length-converted catch-curve (LCCC) method. This occurs because relative age estimates, which are used as a solution to convert the observed sample length composition into ages (the X-axis in the catch-curve plot), will be negatively biased with a larger asymptotic length. Consequently, a much steeper slope (higher mortality estimate) will be produced in the linear regression model (Pauly, 1983). This is simply because it produces an effect where the largest size/age classes of the population, known to always exert a high influence in parameter estimates (Wang et al., 2020), wrongly appear to be missing from the sample and are then assumed by the model to be depleted, while in fact those would not exist. According to our study, errors when converting the length composition of the sample into relative ages derived from erroneous L∞ estimates have a multiplicative effect on the mortality estimation. As such, this can create an undesirable and misleading situation where even an “acceptable” 10% bias in the asymptotic length from the PW method can lead to extreme (>100%) deviations from the “true” total mortality and Z/K values (Fig. 5). Bias in L∞ estimation can also result from sampling-related issues such as selection bias, where, for example, when larger individuals are not well sampled. This could occur in cases where distinct size classes or life stages use different habitats (e.g., larger individuals occupying different depths, Castrejón, Pérez-Castañeda & Defeo, 2005), which would also impact subsequent mortality estimates.

Errors in estimating L∞ can also lead to erroneous estimates of natural mortality (M). This is one of the most important parameters in stock evaluations, but extremely difficult to directly estimate in the absence of pre-exploitation data (Punt et al., 2021). Therefore, stock assessments of data-limited stocks often rely on empirical estimates that either directly incorporate asymptotic length in its equation or use parameters such as the theoretical maximum attainable age, often calculated using the inverse von Bertalanffy equation which uses L∞ (Dureuil et al., 2021; Hewitt & Hoenig, 2005; Jørgensen & Holt, 2013; Kenchington, 2014; Pauly, 1980; Then et al., 2015). It is likely that the natural mortality of the seabob shrimp population in this study, if estimated using outputs from the PW method, would likely be significantly overestimated. According to Punt et al. (2021), positive bias in M can lead to considerable overestimation of stock biomass and a subsequent stumble in its ability to withstand fishing pressure. Ultimately, errors in natural mortality will lead to unreal estimates of fishing mortality if the catch curve method is used, which are crucial input values for models that provide biological reference points for management such as virtual population analysis (VPA) and yield per recruit (YPR) models (Pauly & Soriano, 1986).

Given the absence of information about the “real” parameter distribution for each sample, we acknowledge and emphasize that a central assumption in our real-world study was that outputs from fishboot and length-converted catch-curve methods can be considered reliable reference data for measuring relative bias from the PW-plot outputs. Despite using an unconstrained, robust approach for parameter search in the fishboot, it should be noted that this indirect method (since it does not rely directly rely on length-at-age data to estimate growth) has potential shortcomings that could affect accuracy in parameter estimation such as its sensitivity to the reduced abundance of large individuals in the sample and overall sample size (Schwamborn, Mildenberger & Taylor, 2019). Unconstrained search ranges are recommended for the optimization routines used in TropFishR and fishboot, but current research has demonstrated that the choice of search ranges from wide to narrow impacts parameter estimates (Zhou et al., 2022), possibly because restricting the search space induces the optimization algorithm to get trapped in local maxima (Mildenberger, Taylor & Wolff, 2017). The length-converted catch-curve, used in this study to derive total mortality estimates, has a set of simplifying assumptions that are unlikely to hold on dynamic fished populations such as constant recruitment (steady-state) and mortality rates during the analyzed period, and no variation in selectivity-at-age (Chapman & Robson, 1960). Currently, it is known that mortality greatly varies within a population, usually decreasing with body size due to reduced predation risks (Ramírez-Rodríguez & Arreguín-Sánchez, 2003).

Estimation of natural mortality for invertebrate stocks

In a priori trials and data exploration aimed at conducting a data-limited style stock assessment with the seabob shrimp data, estimation of natural mortality (M) using empirical equations (Then et al., 2015) yielded estimates that were very close (or even above) our estimates for total mortality (Z), which was unexpected. This could indicate, at first sight, a negligible fishing mortality F (assuming Z = M + F). However, considering the intensive trawl fisheries in the study area, a more likely explanation might be a serious inadequacy in the methods used to estimate M, or Z, or both. Since the overall robustness and reliability of the LCCC was verified to be acceptable under most conditions to approximate total mortality rates (Huynh et al., 2018), it is more likely that the estimate of M obtained from empirical equations inferred by relationships between mortality and life-history parameters in fish stocks may not be suitable for penaeid shrimps. Generally, benthic shrimp species spend most of their time buried in sediment, have cryptic, nocturnal habits, and possess long, hard spines (especially X. kroyeri, which has an unusually long rostral spine). Therefore, they may suffer lower predation mortalities than small pelagic fish of similar size (“small pelagics don’t die of old age”). This relationship between morphological features and M has already been demonstrated for fish, where reliable estimates of M are available (Griffiths & Harrod, 2007). In a reanalysis of an updated data set based on the one used for Pauly’s (1980) empirical equation, Griffiths & Harrod (2007) showed that 15 of the 18 species belonging to the 11 families with the lowest mortality rates had morphological defense structures (e.g., venomous spines or defensive armor) or burrowing habits (e.g., in flatfish).

Thus, it is likely that popular empirical equations for estimating M (e.g., Pauly, 1980; Then et al., 2015), which are constructed from fish mortality data and designed for (mostly pelagic and unarmored) fish, are not suitable for well-armored shrimp with distinct behaviors. This is not a widely accepted rationale, except within the study of Mamie et al. (2008). Therefore, we conclude that F cannot be reliably calculated from the difference between Z and empirical M for our data set. This is mainly because our results show that M estimated from empirical equations is virtually equal to Z, which would absurdly imply that there are no fisheries in any of the four study areas, where in fact there are important fisheries targeting seabob shrimp. However, numerous authors have used such empirical equations for many penaeid shrimp stocks (Mehanna, 2000; Leite Jr & Petrere Jr, 2006; Amin et al., 2008; Fernandes, Keunecke & Di Beneditto, 2014; de Barros et al., 2021; Razek et al., 2022), and this approach remains standard for estimating M in data-poor shrimp fisheries. Failures by previous authors to detect issues in M estimation for shrimp stocks might be related to biased Z estimates, which are strongly influenced by L∞ and K during LCCC procedures. In such cases, errors in total mortality estimation could be due to the assumption that L∞ >Lmax, which also leads to biased K estimates if the search space for L∞ is constrained. If we assume that K estimates have been severely biased in previous studies with fixed or constrained L∞ (Schwamborn, 2018, this study), and Z estimates (e.g., from LCCC) are strongly affected by the input K value (Schwamborn, 2018), it is obvious that the Z estimates in these previous studies could have been biased, affecting all subsequent analyses.

Reliable in situ estimates of M for penaeids are scarce (Pérez-Castañeda & Defeo, 2005; Aranceta-Garza et al., 2016). For the brown shrimp Crangon crangon (Caridea: Crangonidae), a small-sized species without a long rostral spine, detailed studies on its mortality were conducted by Oh, Hartnoll & Nash (1999) and Hufnagl et al. (2010), both of which showed a substantially high M/K ratio. Oh, Hartnoll & Nash (1999) found very close estimates for Z (3.96 yr−1) and M (3.60 yr−1), with a Z/K ratio of approximately 4 for the same species in an area with negligible fisheries. Conversely, estimates of M for other penaeids in the adult phase seem to be lower, with a M of about 0.96 year−1 for banana prawn Penaeus merguiensis in tropical Australia (Wang & Haywood, 1999), and around 1.63, 2.52, and 2.89 yr−1 using gnomonic intervals for Farfantepenaeus californiensis, Penaeus stylirostris, and Penaeus vannamei, respectively (Aranceta-Garza et al., 2016). While our data suggests that “empirical” equations used to estimate M may not be appropriate for some crustacean stocks, we recognize our discussion on this issue remains largely speculative, and further studies should examine the relationships between field-based M estimates and life-history/morphological traits.

Estimating Z for data-poor stocks is far from trivial. Severe bias in mortality estimation using common length-based methods has been demonstrated by Hufnagl, Huebert & Temming (2013) and Schwamborn (2018). Obtaining reliable, robust estimates for the mortality components M and F is even more challenging. Ideally, F should be derived from extensive catch and biomass data (Hart, 2012) or tagging (mark and recapture studies) for reliable population modeling and stock assessment. However, tagging shrimp has been proved to be costly, time-consuming, and often problematic, especially due to tagging-related mortality in these more fragile, small organisms (Jewett, 1986). Similarly, catch and biomass time series are rarely available for small-scale artisanal shrimp fisheries. In published shrimp stock assessments, M has generally been generally estimated from empirical equations and used to estimate F from Z. In a few exceptions, M was fixed at an arbitrary value (e.g., 0.3 y−1 or 0.5 y−1) and F was estimated in situ from shrimp catch and biomass data (Mamie et al., 2008; Hart, 2012) using statistical catch-at-age models (Methot Jr & Wetzel, 2013).

In this study, the improved length-converted catch-curve (iLCCC) is of particular novelty. In the old, commonly used LCCC method (Pauly, 1984a; Pauly, 1984b), L∞ necessarily must be above the maximum length. This is not always a realistic assumption. In many cases (e.g., when growth is highly variable and the Z/K ratio is low, as in simulated populations from Schwamborn, 2018), L∞ can be below Lmax, as shown in the present study (also see Schwamborn, 2018; de Barros et al., 2021; Hossain Yeamin & Ohtomi, 2010 for examples of populations with asymptotic length below maximum length). The implicit, un-circumventable rule of the common LCCC that L∞ must be above Lmax can lead to limitations and bias in the choice of L∞ search spaces, thus impeding an unconstrained search. Surprisingly, the serious drawback of the traditional LCCC method has not been adequately highlighted and discussed so far in the published literature. In the present study, we highlight this drawback and present a new, improved iLCCC method (de Barros, 2022), that has no intrinsic limitations regarding the amplitude of a priori defined search spaces for L∞. While our data suggested the iLCCC is adequate to estimate mortality rates, we also highlight the need for more studies to further explore the accuracy of this method against known values for other populations, particularly those with different mortality/growth relationships.

Conclusions

Our case study provided a critical evaluation of widely used traditional length-based methods and demonstrated that they are inappropriate for the study of natural populations, especially in data-poor settings where there is no prior knowledge of the population’s mortality/growth relationship. Unconstrained, robust methods that incorporate uncertainty such as fishboot may be preferable. Improving guidelines and selecting appropriate methods for the assessment of data-poor fish stocks will considerably enhance the prospects of achieving effective management for these resources. Also, considering the recent advancements in the Electronic Length Frequency Analysis (ELEFAN) method, with prompt availability of code-based applications that incorporate multidimensional uncertainty into data-limited stock assessments, the authors call for further methodological research towards the development of new methods and guidelines for data-poor stock assessment.

Supplemental Information

Supplemental Information 1 Raw data

Individual shrimp measurements for the four studied fishing grounds

Supplemental Information 2 Parameter correlations

Scatterplots showing multivariate relationships between different parameter estimates

We are greatly indebted to all members of both the Marine Fish Ecology Laboratory (LEPMAR/UFES) and the Ichthyology Laboratory (Ictiolab/UFES) who helped in the feld and laboratory work.

Additional Information and Declarations

Competing Interests

Author Contributions

Data Availability

The authors declare there are no competing interests.

Matheus de Barros analyzed the data, prepared figures and/or tables, authored or reviewed drafts of the article, and approved the final draft.

Ronaldo Oliveira-Filho conceived and designed the experiments, performed the experiments, authored or reviewed drafts of the article, and approved the final draft.

Alexandre Aschenbrenner conceived and designed the experiments, authored or reviewed drafts of the article, and approved the final draft.

Mauricio Hostim-Silva conceived and designed the experiments, performed the experiments, authored or reviewed drafts of the article, and approved the final draft.

Julien Chiquieri conceived and designed the experiments, performed the experiments, authored or reviewed drafts of the article, and approved the final draft.

Ralf Schwamborn analyzed the data, authored or reviewed drafts of the article, and approved the final draft.

The following information was supplied regarding data availability:

The individual monthly shrimp raw measurements and biological information for each fishing ground are available in the Supplemental Files.

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
