# Peer review of "Evaluation of traditional and bootstrapped methods for assessing data-poor fisheries: a case study on tropical seabob shrimp (Xiphopenaeus kroyeri) with an improved length-based mortality estimation method"

_PeerJ, doi:10.7717/peerj.18397_

## Round 0.1 · original submission · Major Revisions

Please accept my sincere apologies for the exceptional delay in processing your submission. I have now received comments from two reviewers.

The comments are clear and well presented so I will not repeat them here.
I agree with the importance of improving on unreliable stock-assessment methods (e.g. “PW” as stated in the text) and making the more reliable alternatives both more widely available and useable.

I also agree with the reviewers that there are some critical elements missing from your submission, which makes it hard to provide a fair review. Particularly the issues raised by reviewer 2 regarding the need to establish reliability and validity of the new iLCCC method. Based on the reviewers comments I feel that you need to include a simulation exercise as part of the analysis to demonstrate reliability/validity against known plausible values for simulated harvest populations i.e. the aims would have three parts 1) as original.... 2) establishing how valid / reliable the new method (improved length-converted catch-curve - iLCCC) you present is and then to 3) apply to your real world use case of samples of the seabob shrimp. Such a validation exercise has not been carried out previously as far as I can tell (was not part of the de Barros (2022) Brazilian Journal of Biology article you cite, the associated blog address https://rpubs.com/matheusdebarros/907348 returns a 404 error).

I look forward to receiving a revised version that carefully considers all the comments and suggestions. There have been some recent examples presenting new analysis in PEERJ that might help during your review, I list these below.
Hierarchical generalized additive models in ecology: an introduction with mgcv: https://doi.org/10.7717/peerj.6876
PCAtest: testing the statistical significance of Principal Component Analysis in R: https://doi.org/10.7717/peerj.12967
Inventory statistics meet big data: complications for estimating numbers of species: https://doi.org/10.7717/peerj.8872

Reviewer 1 ·

Basic reporting

- I found the article well written overall, with regard to structure and statement of research objectives, methods, and findings. The coauthors use appropriate literature references and provide sufficient field background and context. The article is self-contained with relevant results to the hypothesis.
- There are a number of points where the English could be improved to better conform with standard usage. I have made notations to the draft for potential improvements to the English in a .pdf file which can be provided to the authors. Line numbers where these suggestions are provided include 60-61, 67, 76, 81, 87, 108, 110, 113-114, 140-143, 147-148, 155-156, 175, 177, 285, 288, 364-365, 387, 476-477.
- All appropriate raw data have been made available, and appear consistent with their description in the paper.
- It might be necessary to provide meta data or other additional information to enable other researchers to replicate the results for the data provided.
- Figures seem relevant to the content of the article, of sufficient resolution, and appropriately described and labeled.
- The notation in the version of the article I reviewed could be improved at a number of points, as notated in the .pdf file. Line number references include 38 (subscripts too small), 69 (need subscripts here?), and 232 (confidence limits appear to be misstated).

Experimental design

- The article appears to present original primary research within Aims and Scope of the journal.
- The research question is well defined, relevant & meaningful, and fills an identified knowledge gap regarding the potential bias in traditional estimates of stock assessment parameters in a data poor context. Methods are described with sufficient detail & information to replicate.
- I struggled at points to understand some of the technical details:
* Were time and location of sample collection assumed unnecessary to reflect in the experimental design (line 159)?
* I am quite familiar with bootstrap methodology, but was unclear on meaning of “search algorithm” in the bootstrap context (line 175).
* What does the term “likelihood” mean in the context of this paper (line 191)?
* I was unclear on the definition of “bias” (e.g. line 257) as used in the manuscript, and how exactly it was estimated.
* The authors hint that removals from the population may hamper estimation of length and mortality parameters (line 331), suggesting a problem of “selection bias”. It is not clear whether and how their estimation methodology takes these “missing observations” into account.

Validity of the findings

- Conclusions are well stated, linked to original research question & limited to supporting results, and are valid so far as this reviewer could tell.

Annotated reviews are not available for download in order to protect the identity of reviewers who chose to remain anonymous.

·

Basic reporting

Abstract: The first sentence is stronger than necessary. Perhaps a less general statement would be more appropriate: “Unrealistic model assumptions or improper quantitative methods reduce the reliability of data-limited fisheries assessments.” I am not sure why “indirect routines” is necessary. Please clarify what this means. If both Z and K values are lower for ELEFAN results, please clarify why Z/K values are also lower. “Close to 1” does not provide sufficient clarity here. I do not see any figures or tables that report the values of parameters produced by traditional methods. Please include a table of parameters similar to the table for ELEFAN results or include these parameters in the same table. The improved version of iLCCC is not described in this paper and should not be included in the abstract. Discussion of M is not supported with analysis in this paper, so please remove this from the abstract. Although an unconstrained fishboot routine was used here, there is no sufficient argument against using a constrained routine, so please remove this language from the last sentence.
Introduction: Reword sentence beginning line 60 to be clearer. Perhaps: “Small scale fisheries often lack important data, such as… parameters that can be integrated into stock assessment models more common for industrial fisheries.” Next sentence (line 64): “Therefore, modeling tools were developed to assess data-limited stocks.” Line 68: add semicolons instead of commas so commas can be places before and after t0 and t_anchor. Also, format these terms correctly with subscripts. Line 107: “determinant” doesn’t seem to be the right word here. This sentence is also confusing. Rewording or splitting into two sentences would be a good idea. Line 119: This new method for calculating total mortality is not clear in the text, so either include more detail or remove this second point from the introduction.

Methods: Lines 142-151 seems unnecessary unless the mining disaster has an impact on the analyses presented. In paragraph beginning line 164, please give a brief description of the modified PW plot method. Why were mode+10% and last point 80% max chosen here? These numbers are different than Schwamborn, 2018. Is there a guideline for choosing points according to PW methods? The use of Linf and Z/K should be clearer in this paragraph. Equations to show how Linf and Z/K are input into traditional methods would be helpful. Line 177: please clarify exactly what this rule of thumb is and how 4mm bin size fits the rule of thumb. Beginning line 177, clarify which outputs are the result of TropFishR and which are from fishboot. I don’t see a comparison between the two. Line 188: correct formatting for subscripts. Line 192: This choice of MA is not explained well. Please clarify the procedure and justify the choice of 11. Paragraph 200-212: iLCCC is not presented in this study in any detail, so please do not include it as an aim of this study. It is appropriately referenced, but this is not a publication to present this R package. There are no details about how it performs, its specifications, or even the values generated in fishdynr. Please expound on how M was calculated via Then et al 2015 method. Line 222: was the traditional PW method or the modified Pauly version used? Please be exact.

Results: Line 247: iLCCC method is not described in the methods adequately. Please include a description. Figure 4: PW label misspelled. There is no information on how K is estimated to get Z values from Z/K ratio. K values are unknown, so how can Z be estimated with PW plot? In figure 4, please plot all raw outputs for parameters in separate panels, not only Z.

Discussion: The discussion is quite lengthy for the length of the results. Line 449: Please remove discussion of iLCCC unless adequately presented for evaluation.

Experimental design

Unconstrained parameter searching is not necessarily the most accurate. Zhou et al. 2022 (Shijie Zhou, Trevor Hutton, Yeming Lei, Margaret Miller, Tonya van Der Velde, Roy Aijun Deng, Estimating growth from length frequency distribution: comparison of ELEFAN and Bayesian approaches for red endeavour prawns (Metapenaeus ensis), ICES Journal of Marine Science, Volume 79, Issue 6, August 2022, Pages 1942–1953) demonstrated this choice is sensitive. This search is, in fact, constrained. Please include some sensitivity analysis of results by choice of constraints from wide to narrow (e.g. k: 0.1-3 y^-1 vs k: 0.5-2 y^-1 vs k: 1-1.5 y^-1). Do the overall conclusions of the study depend on this choice? Relative bias equation is wrong. Correct parentheses placement. I assume this is a typographical error and not the formula used to calculate the final results. Line 234: the coefficient of confidence metric should not be used here. Confidence intervals are based on the mean, whereas CC is based on the median. There is no theoretical basis for this as a metric. Please use a more suitable metric like median absolute deviation if you choose to base your results on the median or coefficient of variation if basing results on the mean.

Validity of the findings

These results depend on the reliability of the iLCCC method. As far as I can tell, this method has not been peer reviewed. It is presented as a novel part of this study, but no details are given besides a general description of why it is useful. The ELEFAN results may all be invalid if this iLCCC method is invalid, and there is no way to assess its validity from this manuscript.
Otherwise, I have no concerns for results presented in Figs 1-3. Fig 4 requires plotting of all parameters from both methods. I am not sure how Z was estimated from PW method except by taking k from ELEFAN to estimated from PW-based Z/k. If so, this figure is invalid. Figs 5-6 seem valid.

Additional comments

It is crucial to evaluate PW method in comparison to modern bootstrap and Bayesian approaches. This paper has good intentions to improve the field. Upon inspection of the PW method and its modifications, I agree with the authors that this method is invalid on theoretical grounds. My recommendation for this paper is to either use a case study where Linf exceeds Lmax or to properly explain iLCCC so that it can be evaluated by peer reviewers. Results depend on the validity of this approach. A few other method choices need modification before this study is publishable.

---

## Round 0.2 · Minor Revisions

Thank you for taking the time to provide a much improved version that has addressed the comments from the earlier version. I have now received comments from two reviewers. I agree with reviewer 2 that the text could benefit from a final round of minor corrections to improve flow and clarity (correcting grammatical errors). I also agree that authors could also clarify in the discussion regarding the need for future studies to validate the new method e.g. with additional species. I look forward to receiving your revised version.

Reviewer 1 ·

Basic reporting

I feel the coauthors have addressed the referees' recommended revisions.

Experimental design

Nothing to add to previous comments.

Validity of the findings

Nothing to add to previous comments.

·

Basic reporting

Language and presentation were clear. There are still a few minor grammatical and punctuation errors scattered throughout, so the authors should re-read to correct those they find.

Experimental design

The investigation was much improved in this round, though the conclusions about the improved LCCC being robust are not necessarily true, as there is not independent verification that removing a subset of the data to estimate these parameters should be performed for datasets in general.

Validity of the findings

The iPCCC still needs additional verification, so I would encourage authors to expound on the limits of their conclusions with respect to this method's use. This can be done in the discussion.

---

## Round 0.3 · accepted · Accept

Thank you for your final revisions. The text has been improved following the reviewers suggestions.